# Advanced Removal of Dyes with Tuning Carbon/TiO_2_ Composite Properties

**DOI:** 10.3390/nano14030309

**Published:** 2024-02-03

**Authors:** Halyna Bodnar Yankovych, Coset Abreu-Jaureguí, Judit Farrando-Perez, Inna Melnyk, Miroslava Václavíková, Joaquín Silvestre-Albero

**Affiliations:** 1Institute of Geotechnics, Slovak Academy of Sciences, Watsonova 45, 04001 Košice, Slovakia; in.melnyk@gmail.com (I.M.); vaclavik@saske.sk (M.V.); 2Laboratorio de Materiales Avanzados, Departamento de Química Inorgánica-Instituto Universitario de Materiales, Universidad de Alicante, E-03690 San Vicente del Raspeig, Spain; coset.abreu@ua.es (C.A.-J.); judit.farrando@ua.es (J.F.-P.)

**Keywords:** water treatment, carbon composites, photocatalysis, degradation of organic dyes, textile effluents

## Abstract

This study evaluates the removal of several dyes with different charge properties, i.e., anionic (Acid Red 88), cationic (Basic Red 13), and neutral (Basic Red 5) using transition metal-doped TiO_2_ supported on a high-surface-area activated carbon. Experimental results confirm the successful deposition of TiO_2_ and the derivatives (Zr-, Cu-, and Ce-doped samples) on the surface of the activated carbon material and the development of extended heterojunctions with improved electronic properties. Incorporating a small percentage of dopants significantly improves the adsorption properties of the composites towards the three dyes evaluated, preferentially for sample AC/TiO_2__Zr. Similarly, the photodegradation efficiency highly depends on the nature of the composite evaluated and the characteristics of the dye. Sample AC/TiO_2__Zr demonstrates the best overall removal efficiency for Acid Red 88 and Basic Red 5—83% and 63%, respectively. This promising performance must simultaneously be attributed to a dual mechanism, i.e., adsorption and photodegradation. Notably, the AC/TiO_2__Ce outperformed the other catalysts in eliminating Basic Red 13 (74%/6 h). A possible Acid Red 88 degradation mechanism using AC/TiO_2__Zr was proposed. This study shows that the removal efficiency of AC/TiO_2_ composites strongly depends on both the material and pollutant.

## 1. Introduction

Sustainable water management is one of the 17 Sustainable Development Goals [1] declared by the United Nations in 2015. One of the essential points of these goals is the rational usage of water resources and the development of efficient and eco-friendly wastewater treatment techniques. Xenobiotic pollution considerably affects human beings, ecosystem health, and water supplies and intensifies water stress. Therefore, this issue is still of high concern worldwide. The significant impact on the natural water environment is caused by the discharging of dyes during different anthropogenic activities—biochemistry, medicine, analytical chemistry, and the textile industry [2,3]. Due to the high concentration and stability of dye effluents, conventional physical, chemical, or biological treatment methodologies prove inadequate for efficiently removing substantial volumes of these pollutants. Consequently, developing specialized methods becomes necessary to remove these pollutants from wastewater systems effectively. To overcome these limitations and improve the total purifying process, several treatment techniques can be combined [4].

The application of composites to treat contaminated effluents able to work through a dual adsorptive and photocatalytic mechanism is a very promising and sustainable approach. Among the various types of composite materials, activated carbon-supported titania composites possess several advantages related to their porous structure, elimination potential, and economic usage [5]. Despite the deposition of titania nanoparticles on the activated carbon (AC) surface, the AC substrate retains its high surface area, porous structure, and surface features, which play a crucial role in the adsorption of pollutants on the surface as well as in the subsequent delivery of adsorbate to the photocatalyst [4]. Moreover, these materials exhibit self-cleaning properties and, therefore, can be reused [6]. The adsorptive features of titania-activated carbon composites are especially beneficial for highly concentrated dye wastes since the dye molecules can inhibit the light harvesting by titania and cause photocatalyst poisoning that consequently reduces the photocatalytic performance [7,8]. Since titania is active under ultraviolet irradiation, its application requires special equipment and additional expenses. Therefore, an interesting issue is the modification of titania bandgap by incorporating external dopants, especially transition metals, into its crystal structure or preparation of mixed photocatalytically active oxides that improve the remediation performance of material towards selected pollutants [9]. This can be advantageous in the treatment of highly concentrated textile effluents. Hence, combining activated carbon and titania-based photocatalysts overcomes the limitations related to both materials individually, owing to the formation of highly extended heterojunctions with improved properties.

With this in mind, the goal of this study is to evaluate the performance of composite materials based on granular activated carbon and titania-modified semiconductors for adsorptive and/or photocatalytic removal of three types of dyes: anionic dye—Acid Red 88, cationic dye—Basic Red 13, and neutral dye—Basic Red 5. A series of photocatalysts, in particular, AC/TiO_2_, AC/TiO_2__Ce, AC/TiO_2__Zr, and AC/TiO_2__Cu, were synthesized using the in situ sol-gel method combined with ultrasound treatment [10]. The composites were characterized by advanced analytical tools: SEM-EDS, XRD, XPS, GSA, Raman spectroscopy, and point of zero charge protocol and were used for adsorptive-photocatalytic elimination of selected dyes under ultraviolet irradiation. The kinetics of photocatalysis were processed by a pseudo-first-order model, and the rate constants were estimated. Based on the obtained data, the possible degradation mechanism of Acid Red 88 using photocatalyst AC/TiO_2__Zr was proposed.

## 2. Experimental Section 

### 2.1. Activated Carbon–Titania Composite Materials 

Granular activated carbon (AC) was purchased from ITES Vranov, s.r.o., Dlhé Klčovo, Slovakia, and characterized using nitrogen adsorption at cryogenic temperatures (S_BET_ 1010 m^2^·g^−1^) [11,12]. The composite materials were synthesized by an ultrasound-assisted in situ sol-gel technique described elsewhere [10,13]. For this purpose, 2.5 g of AC was rinsed in isopropanol for 10 min and subsequently air-dried. A sol-gel titania system was prepared as follows: 1.87 g of Ti(IV) isopropoxide (97%), purchased from Sigma Aldrich (Burlington, MA, USA), was put dropwise into 60 mL of isopropanol (Merck absolute, p.a., Darmstadt, Germany) and agitated for 30 min at 1000 rpm. After the complete dissolution of the titania precursor, 1.14 g of concentrated nitric acid (Merck, Darmstadt, Germany) and 12.4 mL of deionized water were dropwise added to the sol system to launch the hydrolysis and gel formation. Then, AC was immersed in a gel-like solution, and the final suspension was treated by ultrasonication for 4 h at room temperature. The raw material was placed in a Petri dish for aging and evaporation of the liquid at room temperature. After drying, the prepared raw material was calcined at 500 °C in an airflow for 5 h. The addition of dopants, namely, Cu(II), Ce(III) and Zr(IV) (in form of Cu(NO_3_)_2_·3H_2_O (p.a., ITES Vranov, s.r.o., Dlhé Klčovo, Slovakia), Ce_2_(CO_3_)_3_ (99.9%, Sigma Aldrich, Burlington, MA, USA), and Zr(IV) isopropoxide (70%, Sigma Aldrich, Burlington, MA, USA), was performed immediately after titania precursor injection. The samples were labeled AC/TiO_2_, AC/TiO_2__Ce_,_ AC/TiO_2__Zr, and AC/TiO_2__Cu. AEROXIDE^©^ TiO_2_ P25 was purchased from Evonik Industries (Essen, Germany) and used as a reference photocatalyst. 

### 2.2. Characterization of Materials 

SEM analysis of composite materials was performed using a high-resolution scanning electron microscope JEOL IT500HR/LA (Musashino, Akishima, Tokyo, Japan) with an EDS detector operated between 0.5 and 30 kV. XPS analysis was carried out with a K-ALPHA Thermo Scientific instrument (Waltham, MA, USA) at a pressure of 5 × 10^−7^ N·m^−2^. The spectra were obtained using Al K radiation (1486.6 eV) with a twin crystal monochromator. X-ray diffraction analysis was performed with a Bruker D8-ADVANCE instrument (Billerica, MA, USA) with a Goebel mirror with a high-temperature chamber and an X-ray generator (KRISTALLOFLEX K 760-80F, Great Malvern, UK) with a copper anode (λ = 1.541838 Å). XRD patterns were recorded between 3° and 80° 2θ with a step of 0.05°. The point of zero charge measurements, called the “11 points experiment”, were conducted according to [14,15]. For this purpose, 50 mg of material was weighed and added to 100 mL of 1 M KCl with an adjusted initial pH (2 to 12) by 0.1 M HNO_3_ and 0.1 M NaOH. The suspensions were agitated in an orbital shaker for 24 h. Afterward, the final pH of each suspension was measured using a portable pH meter HI-8424 HANNA (Smithfield, VA, USA) and the dependence of pH_initial_ = f(pH_final_) was plotted for each sample. The pH_PZC_ was graphically determined at the point when the buffer effect was observed, i.e., the point where the pH does not vary. Raman spectroscopy studies were performed in a Jasco NRS-5100 spectrometer (Easton, MD, USA) working with a laser of 532 nm and a CCD detector (resolution of 6.83 cm^−1^).

### 2.3. Photodegradation Set-Up

The structural formulae of the evaluated dyes are shown in Appendix A. Basic Red 13, BR13 (p.a.), chemical formula C_22_H_26_Cl_2_N_2_, M = 389.36 g·mol^−1^, CAS 3648-36-0, was purchased from Tokyo Chemical Industry CO., LTD. (Tokyo, Japan) Basic Red 5, BR5 (p.a.), chemical formula C_15_H_17_ClN_4_, M = 288.78 g·mol^−1^, CAS 553-24-2, was purchased from Tokyo Chemical Industry CO., LTD. (Tokyo, Japan) and Acid Red 88, AR88 (p.a.), chemical formula C_20_H_13_N_2_NaO_4_S, M = 400.38 g·mol^−1^, CAS 1658-56-6, was purchased from Sigma Aldrich (Burlington, MA, USA). Aqueous solutions were prepared using deionized water. A Jasco V-650 spectrophotometer (Easton, MD, USA) was used for the determination of the dye concentrations in the initial and residual solutions. The maximum absorbance λ_max_ for Basic Red 13 was measured at 523 nm, for Basic Red 5, at 537 nm, and for Acid Red 88, at 505 nm. 

The photocatalytic experiment was carried out in batch mode at room temperature in five-seater photocatalytic reactors (Figure 1). First, 0.05 g of each sample was added to 50 mL of a dye solution with an initial concentration of 200 mg·L^−1^ and an actual pH without adjustment and additional aeration and mixed by a magnetic stirrer for 2 h to reach the equilibrium (“UV off” mode). Then, the lamps (Philips TL-D Blacklight Blue lamps, Eindhoven, The Netherlands) were switched on (“UV on” mode). The aliquots (V = 1 mL) were taken at selected time intervals during the whole experiment to control the changes in dye concentration over time and were centrifuged before UV-Vis analysis. Under the selected conditions, the photolysis experiment under UV light was negligible (not detailed). The adsorption capacity (*q_t_*, mg·g^−1^) and removal efficiency were calculated using Equations (1) and (2), respectively:(1)$$q_{t}=\frac{{(C}_{0}-C_{f})\cdot V}{m}$$
(2)$$R,\%=\frac{C_{0}-C_{t}}{C_{0}}\times 100$$
where *C*_0_ is the initial dye concentration, *C_f_* is the concentration of the dye at the end of the “UV off” experiment, *C_t_* is the dye concentration at the definite time of the photocatalytic reaction (mg·L^−1^), *V* is the volume of the solution (L), and *m* is the mass of the material (g). 

## 3. Results and Discussion

### 3.1. Characteristics of the Synthesized Materials

The surface morphology of the synthesized materials was studied by SEM-EDS technique. As shown in Figure 2, the synthesized samples are quite heterogeneous with a wide particle size distribution. The elemental composition was determined by EDS,. In addition to the presence of carbon and oxygen in the pristine activated carbon, other chemical species such as Ca, Al, Si, and S were identified. Their presence can be explained by the nature of the activated carbon precursor. Concerning the composites, the elemental composition is rather similar among samples, with the additional presence of Ti. The specific quantitative ratio for each composite is reported in Appendix A. The Ti content in the composite materials slightly varies among samples, i.e., 24.2 wt.% in AC/TiO_2_, 22.0 wt.% in AC/TiO_2__Zr, 25.5 wt.% in AC/TiO_2__Ce, and 26.1 wt.% in AC/TiO_2__Cu. In the specific case of the AC/TiO_2__Zr sample, the zirconium content was estimated to be 1.8 wt.%, thus confirming the efficiency of the performed doping. Unfortunately, the availability of the other two elements, Ce and Cu, in the AC/TiO_2__Ce and AC/TiO_2__Cu samples, respectively, could not be confirmed. Most probably, these two elements were incorporated in the titania lattice and, therefore, could not be detected by SEM. 

Raman spectrometry was applied to the synthesized samples to identify the main vibrational modes within the composites (Figure 3). The Raman spectrum of pristine activated carbon shows the two characteristic peaks at 1336 cm^−1^ and 1588 cm^−1^, corresponding to the D (sp^3^) and G (sp^2^) contributions, respectively. The presence of 2D and S3 peaks on the activated carbon Raman spectrum evidences the layered packing structure in these carbon materials [16]. Additionally, the ratio between intensities I_D_/I_G_ determines the graphitization degree. This value is 1.06 for the pristine activated carbon [16]. The Raman spectrum of the AC/TiO_2_ sample shows the characteristic D and G bands of the carbon and the characteristic peaks of TiO_2_ at 152 cm^−1^, 391 cm^−1^, 493 cm^−1^, and 624 cm^−1^ [17]. The peaks can be attributed to a mixture of anatase, brookite, and rutile (at 152 and 624 cm^−1^), anatase/brookite (at 391 cm^−1^), and anatase at 493 cm^−1^. Thus, the evaluated sample contains a mixture of three titania polymorphs. In the specific case of the doped samples, i.e., AC/TiO_2__Zr, AC/TiO_2__Ce, and AC/TiO_2__Cu, some shifts in the bands attributed to titania were noticed, together with the disappearance of the 2D and S3 peaks; however, the presence of all polymorphs was still observed. The mentioned observations could be attributed to the modification of the C-C vibrational modes upon modification with TiO_2_ and/or doped TiO_2_ but without modification of the graphitization degree. At this point, it is important to highlight that the Ti-O vibrational mode E_g_(1) at 152 cm^−1^ remains unchanged.

X-ray diffraction patterns were used to investigate the crystalline structure of the synthesized materials. As demonstrated in Figure 4, the pristine activated carbon exhibits the characteristic peaks of amorphous carbon with broad peaks at 2θ: 22° and 44° and peaks attributed to quartz ***Q***, with the most intensive patterns at 2θ: 20.9°, 26.7°, and 50.2° (ICDD Card No. 00-046-1045). The XRD pattern of the composites shows the contributions from anatase (ICDD Card No. 21-1272), brookite (ICDD Card No. 29-1360), and rutile (ICDD Card No. 21-1276) with XRD patterns of anatase **A** at 2θ: 25.23° (101), 37.72° (004), and 47.89° (200); rutile **R** at 2θ: 27.45° (110), 36.10° (101), 41.26° (111), and 54.35° (211); and brookite **B** at 2 θ: 25.35° (210), 25.71° (111), 30.83° (211), and 36.29° (102). The results coincide with the data from Raman spectroscopy, where the same composition of synthesized samples was detected. Overall, the XRD peaks due to TiO_2_ cannot be properly resolved due to the small TiO_2_ content in the composites and the proper dispersion (only some broad contributions due to anatase can be appreciated), while no sign for the dopants could be observed. It is assumed that these dopants could be integrated into the titania lattice without large changes in the titania crystalline structure [18]. At this point, it is important to highlight that the presence of anatase, brookite, or rutile in the composites can boost photocatalytic performance, particularly in the removal of non-biodegradable persistent pollutants [19]. 

The assessment of the X-ray photoelectron spectra of pristine AC and the synthesized composites is summarized in Table 1. The characteristic C1s peak at 284.6 eV binding energy in pristine AC corresponds to C-C bonds in graphite microdomains. Additional peaks in the C1s spectra at higher binding energies—285.8 eV, 287.9 eV, and 290.5 eV—are related to different types of C-O bonds, namely, C-O, C=O, and O-C=O, respectively. The XPS spectrum of the O1s core level contains overlapping peaks with binding energies at 531.0 eV (C=O), 532.7 eV (C-O, OH), and 534.3 eV (O-C=O). Additionally, the signals of Al2p (75.3 eV), S2p (163.8 eV), and Si2p (103.5 eV) were registered during the analysis. The presence of these elements in pristine AC is in close agreement with the SEM-EDS analysis results described above and can be explained by the nature of the AC precursor. For the composites, two characteristic peaks of Ti2p at 459.4 eV (Ti2p_3/2_) and 465.2 eV (Ti2p_1/2_) appeared on the AC/TiO_2_ XPS spectrum, thus providing clear evidence about the presence of Ti^4+^ in these samples [20]. The occurrence of an additional overlapping peak in the O1s spectrum at 530.5 eV, mainly for the composites, can be assigned to the formation of Ti-O bonds between titania and the carbon surface from the supports, thus confirming the success in the developed carbon–oxide heterojunction. A similar feature was observed for the other three composite samples (see Table 1).

The presence of doping elements was examined as well. As can be seen, zirconium was successfully detected on the XPS spectrum of sample AC/TiO_2__Zr with a contribution at 183.4 eV (Zr3d) and a concentration level of 0.3 at.% (Appendix A). This binding energy corresponds to Zr^4+^ species, probably in the form of ZrO_2_ and/or ZrSiO_4_ [20]. The formation of zirconium silicate can be explained by the presence of silica in the composition of the pristine AC. It is interesting to note that ZrSiO_4_ possesses photocatalytic features and excellent adsorption properties [21]. Therefore, it is expected that AC/TiO_2__Zr will have improved photocatalytic characteristics. Notably, the XPS spectra of AC/TiO_2__Ce and AC/TiO_2__Cu do not possess any peaks attributed to Ce3d and Cu2p. These dopant metals were not detected. This fact can be interpreted by the absence of these elements on the composite’s surface or their very low content that cannot be detected by the analyzer. 

The changes in surface chemistry were examined by the “11 points experiment” [15]. The values of the point of zero charge were estimated for pristine AC pH_pzc_ = 6.8, AC/TiO_2_—6.4, AC/TiO_2__Zr—6.5, AC/TiO_2__Ce—6.4, and AC/TiO_2__Cu—8.0. It is noteworthy that the modification of AC with titanium dioxide-doped Cu drastically shifted the surface charge of the activated carbon carrier to a more basic nature. This fact can be explained by the increased nitrogen content on the sample surface calculated from XPS data (Appendix A), which was twice as high as in other materials—1.4 at.%. The amount of protonated amino groups in AC/TiO_2__Cu was 0.8 at.%, while the other samples had near 0.4 at.%. Therefore, it can be assumed that these moieties contribute to the higher surface basicity of the AC/TiO_2__Cu sample. 

Textural properties of the synthesized composites were characterized by N_2_ adsorption–desorption isotherms at cryogenic temperature. Adsorption data were used to estimate the BET surface area (Appendix A and Table 2). According to Table 2, the surface area of the materials is quite high: 1010 m^2^·g^−1^ for AC, 757 m^2^·g^−1^ for AC/TiO_2_, 720 m^2^·g^−1^ for AC/TiO_2__Ce, 672 m^2^·g^−1^ for AC/TiO_2__Zr, and 756 m^2^·g^−1^ for AC/TiO_2__Cu. The nitrogen adsorption isotherms for the composites exhibit type IV, attributed to mesoporous materials, with an additional H4 hysteresis loop [22]. Total pore volume (V_tot_) was slightly reduced in the composites, i.e., AC/TiO_2_, AC/TiO_2__Zr, and AC/TiO_2__Ce, compared with pristine AC. However, in the specific case of sample AC/TiO_2__Cu, the V_tot_ was higher than in pristine AC. It is suggested that copper(II) nitrate, which was used as a precursor of Cu in the doping procedure (Section 2.1), probably promoted condensation in the intergranular space of AC, and mesopores were formed in these voids [23]. Interestingly, the micropore volume values decreased for all composites compared with AC; therefore, it is assumed that the photocatalytic species were deposited in the microporous network. To sum up, the modification of AC with titania semiconductors influenced the textural properties of activated carbon, while the incorporation of dopants has a small effect on the final textural properties of the composites.

### 3.2. Adsorptive–Photocatalytic Properties of Synthesized Materials towards Selected Dyes

The photocatalytic performance of the studied materials towards the three selected dyes was examined using UV light. The photodegradation curves are presented in Figure 5A–C. For a better identification of the photocatalytic efficiency of the composites, additional photodegradation tests were performed in the presence of the pristine granular activated carbon and P25 (AEROXIDE^©^) as reference materials. The initial concentration of each dye was always 200 mg·L^−1^.

Figure 5A clearly shows that the removal capacity for anionic dye AR88 (pK_a_ = 10.7) in the “UV off” regime (in the dark mode) is quite different among samples. The composites showed a much faster removal of the dye in comparison with the pristine AC, i.e., while the pristine activated carbon eliminates nearly 20% of AR88 within 120 min, the composites removed ca. 40–60% of the initial dye concentration in the same time interval. These observations confirm titania’s significant impact on the composite samples’ adsorptive properties. The best-performing material is the AC/TiO_2__Zr sample. The removal of dye in the dark mode is exclusively due to the adsorptive interactions with the composite samples. As shown in Figure 5D, the amount of AR88 adsorbed is enhanced compared with pristine AC (q_t_(AC) = 67.0 mg·g^−1^). More specifically, the adsorption capacity for AC/TiO_2_, AC/TiO_2__Ce, and AC/TiO_2__Cu is similar among samples and in the range of 125.0–140.0 mg·g^−1^, whereas for AC/TiO_2__Zr, this value reaches 167.0 mg·g^−1^. At this point, it is interesting to highlight that the increase in the adsorption capacity does not follow a clear correlation with the BET surface area, thus anticipating the presence of additional interaction processes (e.g., through surface defects) to explain the observed adsorption performance. Indeed, a good agreement between the O/C ratio (Table 3) and the adsorption capacity is observable, i.e., an enhancement in the O/C ratio causes an increase in the amount of adsorbed dye. A higher O/C ratio indicates an improved oxygen surface chemistry; these oxygen functionalities play a crucial role in the removal of AR88 through specific interactions. Additionally, the pH for all samples analyzed (pH_pzc_) is higher than the actual pH of the evaluated Acid Red 88 solution (pH = 6.0); therefore, the surface of composites and activated carbon bears a positive charge. Since AR88 is an anionic dye, electrostatic interactions can occur between the dye molecules and the composites’ surfaces. Regarding the commercial titania P25, the material did not possess adsorptive properties towards AR88; therefore, it is expected that only photodegradation will be involved in the total removal process. After switching on the irradiation (“UV on” mode), a small decrease in relative concentrations was observed for all samples. For instance, for the best performing AC/TiO_2__Zr, this value reached 24% removal. However, AC/TiO_2__Cu and pristine AC showed relatively higher values of 39% and 36% removal, respectively. P25 possessed the best photocatalytic performance in “UV on” mode, i.e., 56% removal. The total removal efficiency of the materials in dark and light conditions was: P25 56% < AC 62% < AC/TiO_2_ 72% < AC/TiO_2__Zr 83% < AC/TiO_2__Ce 78% < AC/TiO_2__Cu 83% (Table 3).

The photocatalytic behavior of cationic BR13 (pKa = 2.9) was slightly different from AR88. Figure 5B illustrates the decolorization kinetics for all samples evaluated. The best adsorption performance towards BR13 in the “dark mode” was established for the AC/TiO_2_ and AC/TiO_2__Zr composites with 47% (q_t_ = 97.9 mg·g^−1^) and 43% (q_t_ = 87.1 mg·g^−1^) removal, respectively. These results can be explained by the electrostatic interactions between the negatively charged composites’ surfaces and the positively charged dye molecules. Indeed, under the evaluated adsorption conditions, BR13 molecules are charged (the initial pH of the dye solution was 6.5). It is interesting to highlight that the pristine AC showed 16.2 mg·g^−1^ adsorption capacity, which is 8% of the total initial dye concentration. Samples AC/TiO_2__Ce and AC/TiO_2__Cu had similar values for q_t_. Under the “UV on” mode, the trend in photocatalytic performance was reversed: almost all materials with high adsorption affinity had reduced photocatalytic efficiency, and vice versa, which coincides with previous findings [24]. Thus, pristine AC demonstrated a 39% concentration decrease, whereas AC/TiO_2_ exhibited only 11% efficiency, AC/TiO_2__Zr had 18%, and AC/TiO_2__Cu performed a 12% concentration reduction for BR13. It is noteworthy that AC/TiO_2__Ce retained high photocatalytic performance even after the adsorption of BR13. Pristine titania P25 showed 75% efficiency. 

Basic Red 5 was chosen as a representative dye with a neutral nature (pK_a_ = 6.7, 7.4). The kinetic curves of BR5 are illustrated in Figure 5C. The results indicated that the decolorization of BR5 by the evaluated materials achieved a total efficiency of up to 63%. The best-performing material was AC/TiO_2__Zr, with up to 42% removal efficiency through adsorption and 21% removal efficiency through photocatalysis. It should be pointed out that the pristine AC and AC/TiO_2__Ce samples possessed the highest photocatalytic efficiency among the other materials, i.e., 30% and 27%, respectively. Reduced photocatalytic decomposition was observed for sample AC/TiO_2__Cu with only 6% removal efficiency, although the adsorption was relatively high—67.8 mg·g^−1^. Commercial titania showed a 44% reduction in the initial BR5 concentration. All carbon-based materials demonstrated an increase in dye concentration after 240 min of photocatalytic treatment, which can be explained by the release of adsorbed dye molecules caused by a shift in adsorption–desorption–decomposition equilibrium: the desorption rate was higher than the decomposition rate. BR5 showed the highest stability towards photocatalytic treatment, which can be explained by its neutral nature. Under selected conditions, only about 15% of the dye molecules were charged, and 85% remained unchanged, which made the conjugated dye system less sensitive to ultraviolet light and, therefore, photodegradation.

### 3.3. Kinetics Modeling

The photocatalytic kinetic curves (“UV on” mode kinetics) were processed by a pseudo-first-order kinetic model, and the rate constants were calculated from a linear range of ln(C/C_0_) = f(t) dependences. The observed rate constants range from 0.0006 to 0.0064 min^−1^, which depends on the nature of the material and dye (Table 4). It is interesting to note that bare activated carbon showed surprisingly high values of rate constants for all the dyes and is competitive with the composites. The highest rate constants for AR88 were observed for AC/TiO_2__Cu k = 0.0041 min^−1^ and P25 k = 0.0038 min^−1^. The rate constants of BR13 had different values, with the highest rate for P25 0.0056 min^−1^. Among the composite materials, the highest rate constant was determined for AC/TiO_2__Ce k = 0.0030 min^−1^. Moreover, AC/TiO_2__Ce possessed the highest removal rate for BR5—0.0064 min^−1^.

### 3.4. Proposed Degradation Mechanism

The obtained results allow us to propose the possible photodecomposition mechanism for AR88 using AC/TiO_2__Zr as a photocatalyst (Figure 6). According to [25], the band gap of activated carbon is below 4 eV, for ZrO_2_—5.1 eV [26] and for TiO_2_—3.2 eV [10]. Since these materials are in interfacial contact, the electron transfer occurs between energy levels of distinct band gaps of these solid phases. ZrO_2_ has the largest band gap; therefore, the photogenerated electrons from the conduction band (CB) of ZrO_2_ can move to the CB of activated carbon and/or TiO_2_ and participate in the generation of superoxide radicals, which further become involved in the decomposition of AR88 molecules. The excited electrons from the CB of activated carbon can migrate to the CB of titania and produce superoxide radicals. In turn, the resulting electron holes on titania’s valence band (VB) move to the VB of activated carbon and/or zirconia dioxide while generating hydroxide radicals from water molecules. 

## 4. Conclusions

Four composite materials based on granular activated carbon and titania semiconductor materials were synthesized and examined for photocatalytic removal of Acid Red 88, Basic Red 13, and Basic Red 5 under ultraviolet irradiation. The composites were characterized by SEM-EDS, XRD, XPS, GSA, Raman spectroscopy, and point of zero charge measurements. The functionalization of activated carbon with titania materials influenced its textural properties and surface chemistry and improved the adsorptive-photocatalytic features of synthesized materials towards the selected dyes. Composites possessed higher values of adsorption capacities in comparison with bare activated carbon: up to 167 mg·g^−1^, 100 mg·g^−1^, and 80 mg·g^−1^ for Acid Red 88, Basic Red 13, and Basic Red 5, respectively. The estimated rate constants of the photocatalytic kinetic curves ranged from 0.0006 to 0.0064 min^−1^ and were related to both the material and the dye. Bare activated carbon demonstrated competitive behavior in the photodecomposition of dyes. The best overall removal performance for Acid Red 88 and Basic Red 5 was exhibited by AC/TiO_2__Zr—83% and 63%, respectively. AC/TiO_2__Ce had the highest efficiency in eliminating Basic Red 13—74% for 6 h. Based on the obtained data, the possible mechanism of Acid Red 88 degradation using AC/TiO_2__Zr was proposed.

## Figures and Tables

**Figure 1 nanomaterials-14-00309-f001:**
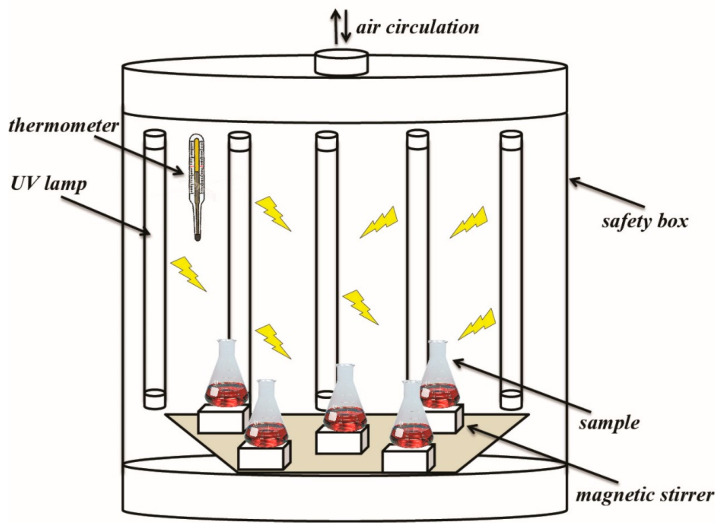
Scheme of the used photocatalytic reactor.

**Figure 2 nanomaterials-14-00309-f002:**
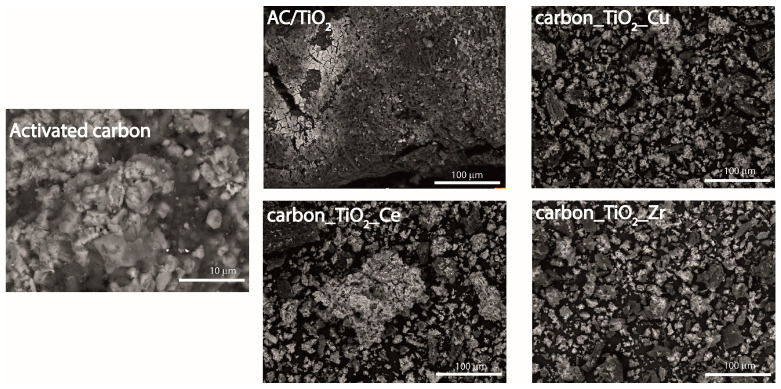
SEM images of bare activated carbon and synthesized materials.

**Figure 3 nanomaterials-14-00309-f003:**
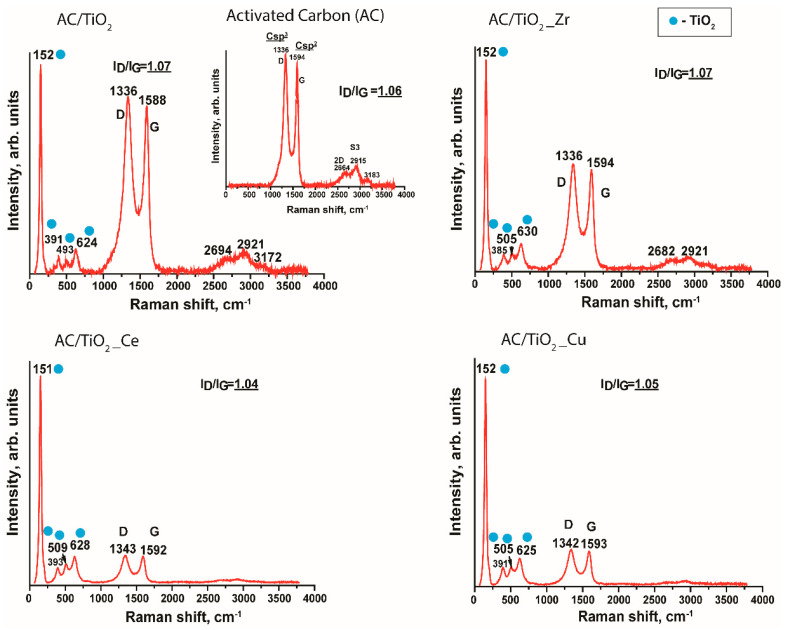
Raman spectra of pristine activated carbon and synthesized composites.

**Figure 4 nanomaterials-14-00309-f004:**
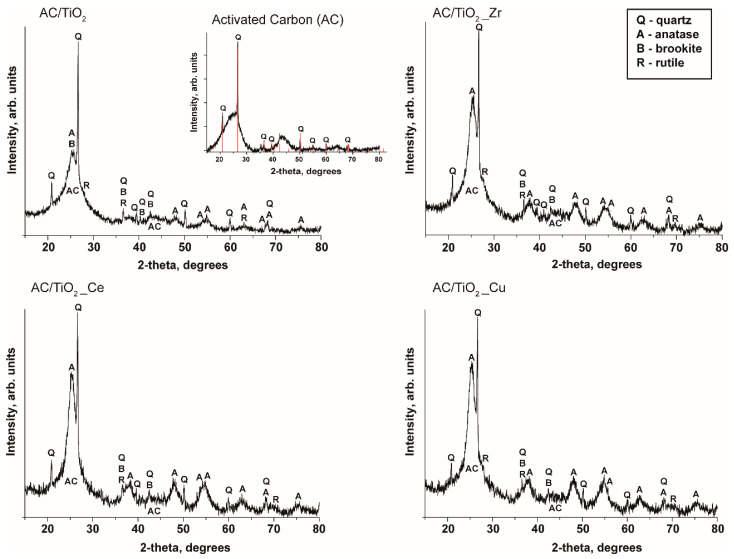
XRD patterns of bare granular activated carbon and synthesized composites (**Q**—quartz, **A**—anatase, **B**—brookite, **R**—rutile).

**Figure 5 nanomaterials-14-00309-f005:**
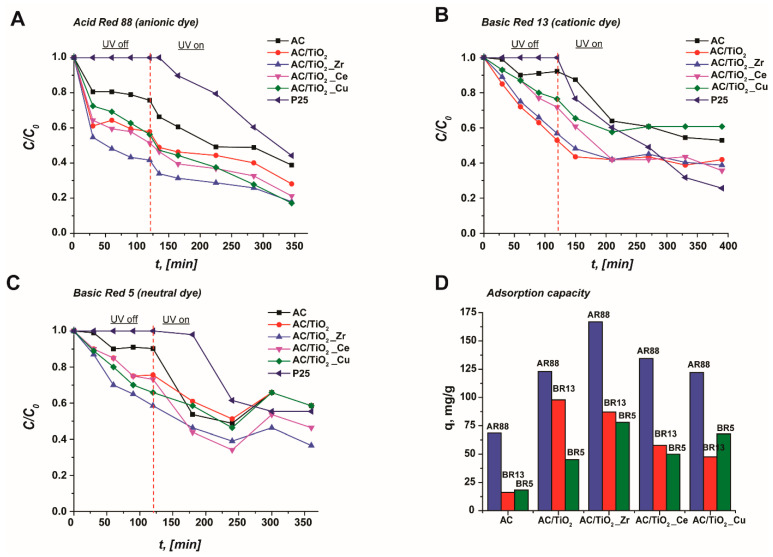
Photodegradation kinetic curves of target dyes (**A**–**C**) and calculated adsorption capacities (**D**) of used materials (AR88—Acid Red 88, BR13—Basic Red 13, BR5—Basic Red 5; conditions—C_dye_ = 200 mg·L^−1^, pH—the initial pH of dye solution without adjustment, C_catalyst_ = 1 g·L^−1^).

**Figure 6 nanomaterials-14-00309-f006:**
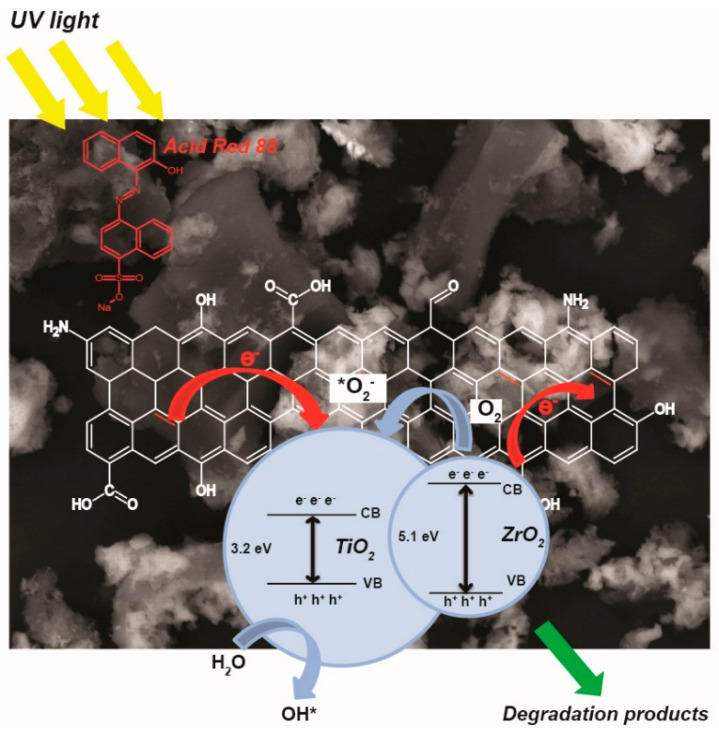
Possible electron transfer mechanism of the most performed composite in AR88 decomposition.

**Table 1 nanomaterials-14-00309-t001:** The summary of XPS analysis of the synthesized composites.

Sample	C1s	O1s	N1s	S2p	Ti2p and Dopants
AC	284.6 eV C-C, C-H 285.8 eV C-O 287.9 eV C=O 290.5 eV O-C=O	531.0 eV C=O 532.7 eV C-O, anhydride/lactone 534.3 eV carboxyl	-	163.8 eV R-SH	-
AC/TiO_2_	285.0 eV C-C, C-H 286.1 eV C-O 288.3 eV C=O 290.8 eV O-C=O	530.5 eV Ti-O 531.4 eV C=O 533.1 eV C-O, anhydride/lactone 534.8 eV carboxyl	398.8 eV NH_2_ 400.4 eV NH_3_^+^ 401.9 eV NO_3_^-^	163.5 eV R-SH 169.3 eV sulfate	460.0 eV (Ti^4+^)
AC/TiO_2__Zr	285.0 eV C-C, C-H 286.2 eV C-O 288.4 eV C=O 290.8 eV O-C=O	530.5 eV Ti-O 531.3 eV C=O 532.7 eV C-O, anhydride/lactone 534.2 eV carboxyl	398.6 eV NH_2_ 400.2 eV NH_3_^+^ 401.8 eV NO_3_^-^	163.8 eV R-SH 169.2 eV sulfate	459.9 eV (Ti^4+^) Zr3d: 183.4 eV ZrO_2_ and/or ZrSiO_4_
AC/TiO_2__Ce	285.0 eV C-C, C-H 285.9 eV C-O 287.0 eV C=O 289.3 eV O-C=O	530.5 eV Ti-O 531.2 eV C=O 533.0 eV C-O, anhydride/lactone 534.4 eV carboxyl	398.7 eV NH_2_ 399.9 eV NH_3_^+^ 401.5 eV NO_3_^-^	163.7 eV R-SH 169.1 eV sulphate	459.8 eV (Ti^4+^) Ce3d ND
AC/TiO_2__Cu	285.0 eV C-C, C-H 287.0 eV C-O 289.1 eV C=O 291.2 eV O-C=O	530.4 eV Ti-O 531.1 eV C=O 532.9 eV C-O, anhydride/lactone 534.0 eV carboxyl	398.6 eV NH_2_ 400.2 eV NH_3_^+^ 401.7 eV NO_3_^-^	163.8 eV R-SH 168.8 eV sulphate	459.5 eV (Ti^4+^) Cu2p ND

ND—not detected.

**Table 2 nanomaterials-14-00309-t002:** Surface area and porosity of the synthesized composites.

Sample	S_BET_, m^2^·g^−1^	V_tot_, cm^3^·g^−1^	V_micro_, cm^3^·g^−1^	D_pore_, nm
AC	1010	0.61	0.36	4.0
AC/TiO_2_	757	0.54	0.14	3.8
AC/TiO_2__Zr	672	0.57	0.14	3.8
AC/TiO_2__Ce	720	0.51	0.20	3.8
AC/TiO_2__Cu	756	0.76	0.10	3.8

**Table 3 nanomaterials-14-00309-t003:** Summary of the total removal efficiency of the studied materials.

Sample		AC	AC/TiO_2_	AC/TiO_2__Zr	AC/TiO_2__Ce	AC/TiO_2__Cu
O/C ratio (from EDS data)		0.21	0.97	1.30	1.24	1.07
pH_PZC_		6.8	6.4	6.5	6.4	8.0
AR88 *, in %	Adsorption	24	42	58	48	44
	Photocatalysis	36	30	25	30	39
	Total	60	72	83	78	83
BR13 *, in %	Adsorption	8	47	43	38	34
	Photocatalysis	39	11	18	36	12
	Total	47	58	61	74	46
BR5 *, in %	Adsorption	10	25	42	27	34
	Photocatalysis	30	16	21	27	6
	Total	40	41	63	54	40

*—initial dye concentration 200 mg·L^−1^.

**Table 4 nanomaterials-14-00309-t004:** Comparison of k values for the studied materials of the selected dyes.

Compound	AR88	R^2^	BR13	R^2^	BR5	R^2^
AC	0.0028	0.99	0.0022	0.96	0.0018	0.99
AC/TiO_2_	0.0012	0.97	0.0006	0.99	0.0010	0.95
AC/TiO_2__Zr	0.0017	0.99	0.0009	0.94	0.0014	0.96
AC/TiO_2__Ce	0.0028	0.98	0.0030	0.99	0.0064	0.96
AC/TiO_2__Cu	0.0041	0.99	0.0024	0.99	0.0029	0.96
P25	0.0038	0.99	0.0056	0.98	0.0009	0.89

## Data Availability

Data are contained within the article and Appendix A.

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
