# Peer review of "Advanced Removal of Dyes with Tuning Carbon/TiO2 Composite Properties"

_nanomaterials, 2024, doi:10.3390/nano14030309_

Round 1

Reviewer 1 Report

Comments and Suggestions for Authors

The photocatalytic removal activity over several dyes was measured by using several metal-doped titanium dioxide loaded on activated carbon. The microstructure, chemical composition  as well as surface properties were analyzed.  However, it seems that the doping modification had little effect on the photocatalytic removal of these three dyes.  The activity curves of different catalysts were so entangled that they appeared to be within the margin of errors.  In this way, the significance of composites creation in such a complex way was not very strong. In addition, this paper still lacked some importatnt contents, such as band structure analyses, active species capture, and interface carriers migration, etc. Thus, I had to reject this work for publication.

Author Response

Answer: We would like to thank the reviewer for the time used to review our manuscript. We apologize that our manuscript did not meet his/her expectations and we were unable to convince him about the quality of the manuscript. The main goal of the manuscript was i) to compare the photocatalytic performance of a series of TiO2-based materials, ii) to emphasize the promoting role of carbon as an additive, and iii) to demonstrate how sensitive are the photocatalysts to the nature of the dye to be degraded. Although it is true that the experimental results are not outstanding, we believe that the information provided can guide future studies about the three points raised above (e.g., effect of photocatalyst, effect of carbon as additive, and effect of the dye to be degraded). 

Reviewer 2 Report

Comments and Suggestions for Authors

Four composite materials based on granular activated carbon and titania semiconductor materials were synthesized and examined for photocatalytic removal of Acid Red 88, Basic Red 13 and Basic Red 5 under ultraviolet irradiation. Experimental results confirm that sample AC/TiO 2 _Zr demonstrates the best overall removal efficiency for Acid Red 88 and Basic Red 5 – 83% and 63%, respectively. I agree to publish this paper after minor revision.

Following question need explanation further:

1.     XRD (FIG, 4) indicate there are quartz in samples. Where is come from?

2.     The “heterojunctions “ was mentioned in ABSTRACT. How to verify its existence? Please provide evidence.

Author Response

Four composite materials based on granular activated carbon and titania semiconductor materials were synthesized and examined for photocatalytic removal of Acid Red 88, Basic Red 13 and Basic Red 5 under ultraviolet irradiation. Experimental results confirm that sample AC/TiO2_Zr demonstrates the best overall removal efficiency for Acid Red 88 and Basic Red 5 – 83% and 63%, respectively. I agree to publish this paper after minor revision. 

Following question need explanation further: 

  1. XRD (FIG, 4) indicate there are quartz in samples. Where is come from? 

Answer: We would like to thank the reviewer for the nice question. The presence of quartz in the synthesized samples can be explained by the nature of the carbon precursor used, which was used as a carrier for the photocatalysts. The results coincide with SEM-EDS analysis that also revealed Si in the activated carbon structure and further in synthesized composites. The availability of quartz in activated carbon composition has been also observed in other published manuscripts:

  • Dutournié, P., Bruneau, M., Brendlé, J., Limousy, L., & Pluchon, S. (2019). Mass transfer modelling in clay-based material: Estimation of apparent diffusivity of a molecule of interest. Comptes Rendus Chimie, 22(2-3), 250-257.
  • Silva, T. L., Ronix, A., Pezoti, O., Souza, L. S., Leandro, P. K., Bedin, K. C., Almeida, V. C. (2016). Mesoporous activated carbon from industrial laundry sewage sludge: Adsorption studies of reactive dye Remazol Brilliant Blue R. Chemical Engineering Journal, 303, 467-476.
  • Li, W. H., Yue, Q. Y., Gao, B. Y., Ma, Z. H., Li, Y. J., & Zhao, H. X. (2011). Preparation and utilization of sludge-based activated carbon for the adsorption of dyes from aqueous solutions. Chemical engineering journal, 171(1), 320-327.

  1. The “heterojunctions“ was mentioned in ABSTRACT. How to verify its existence? Please provide evidence.

Answer: Thank You for the question. A heterojunction is a type of interface interaction between different semiconducting materials that have different bandgap energies. In photocatalysis, heterojunctions are used to improve the efficiency of the photocatalytic process by increasing the separation of photo-generated electron-hole pairs, which reduces the recombination rate and enhances the photocatalytic activity [1]. The strategy for the construction of heterojunction by coupling TiO2 semiconductor with a secondary substance including carbon-based materials and/or other semiconductors is widely applied. For example, the evaluated AC/TiO2_Zr photocatalytic system is complex and consists of three titania polymorphs (anatase, brookite and rutile), zirconium oxide and activated carbon. The detailed analysis of sample composition by XRD and Raman spectroscopy confirmed the presence of three titania polymorphs in the synthesized material. It is known that an anatase-rutile-brookite system can form a titania-based phase heterojunction or facet heterojunction [2,3]. Other important factor influences the heterojunction is close contact between the composite units, which in this case are titania, zirconium oxide and activated carbon. SEM-EDS and XPS analysis verified the presence of zirconium oxide along with titanium oxide on the material’s surface and their near-uniform distribution. The impact of activated carbon (which is known to possess conducting properties) on the system consists in a synergistic effect caused by the textural properties of AC [4]; elimination of the inhibitory effect of contaminant concentration on photocatalytic activity of photocatalyst [5]; levelling of mass transfer limitations [6]; prevention of agglomeration of semiconductor particles that causes the deactivation of photocatalyst [7]. Based on the mentioned above, it can be suggested that the following electron transfer mechanism can exist in evaluated sample. Since the band gap of activated carbon is below 4 eV, for ZrO2 – 5.1 eV [8] and TiO2 – 3.2 eV [9] and interfacial contact is present between composite components, the electron transfer occurs between energy levels of distinct band gaps of these solid phases. Thus, ZrO2 has the largest band gap the photogenerated electrons from the conduction band (CB) of ZrO2 can move to the CB of activated carbon and/or TiO2 and participate in the generation of superoxide radicals, which further become involved in the decomposition of dye molecules. The excited electrons from the CB of activated carbon can migrate to the CB of titania and produce superoxide radicals. In turn, the resulting electron holes on titania’s valence band (VB) move to the VB of activated carbon and/or zirconia dioxide while generating hydroxide radicals from water molecules. This system was the best performed in most of cases, which was verified by the high rate constants.

Ref.:

  1. Balapure, A., Dutta, J. R., & Ganesan, R. (2023). Recent advances in semiconductor heterojunction: a detailed review of fundamentals of the photocatalysis, charge transfer mechanism, and materials. RSC Applied Interfaces.
  2. Wei, H., McMaster, W. A., Tan, J. Z., Chen, D., & Caruso, R. A. (2018). Tricomponent brookite/anatase TiO2/g-C3N4 heterojunction in mesoporous hollow microspheres for enhanced visible-light photocatalysis. Journal of Materials Chemistry A, 6(16), 7236-7245;
  3. Li, K., Teng, C., Wang, S., & Min, Q. (2021). Recent advances in TiO2-based heterojunctions for photocatalytic CO2 reduction with water oxidation: A review. Frontiers in Chemistry, 9, 637501
  4. Xue, G., Liu, H., Chen, Q., Hills, C., Tyrer, M., & Innocent, F. (2011). Synergy between surface adsorption and photocatalysis during degradation of humic acid on TiO2/activated carbon composites. Journal of Hazardous Materials, 186(1), 765-772.
  5. Asenjo, N. G., Santamaria, R., Blanco, C., Granda, M., Alvarez, P., & Menendez, R. (2013). Correct use of the Langmuir–Hinshelwood equation for proving the absence of a synergy effect in the photocatalytic degradation of phenol on a suspended mixture of titania and activated carbon. Carbon, 55, 62-69.
  6. Ghasemi, B., Anvaripour, B., Jorfi, S., & Jaafarzadeh, N. (2016). Enhanced photocatalytic degradation and mineralization of furfural using UVC/TiO2/GAC composite in aqueous solution. International Journal of Photoenergy, 2016.
  7. Andrade, M. A., Carmona, R. J., Mestre, A. S., Matos, J., Carvalho, A. P., & Ania, C. O. (2014). Visible light driven photooxidation of phenol on TiO2/Cu-loaded carbon catalysts. Carbon, 76, 183-192.
  8. Fakhri, A.; Behrouz, S.; Tyagi, I.; Agarwal, S.; Gupta, V.K. Synthesis and Characterization of ZrO2 and Carbon-Doped ZrO2 Nanoparticles for Photocatalytic Application. J. Mol. Liq. 2016, 216, 342–346, doi:10.1016/j.molliq.2016.01.046.
  9. Yankovych, H.; Bodnár, G.; Elsaesser, M.S.; Fizer, M.; Storozhuk, L.; Kolev, H.; Melnyk, I.; Václavíková, M. Carbon Composites for Rapid and Effective Photodegradation of 4-Halogenophenols: Characterization, Removal Performance, and Computational Studies. J. Photochem. Photobiol. Chem. 2023, 441, 114753, doi:10.1016/j.jphotochem.2023.114753.

Reviewer 3 Report

Comments and Suggestions for Authors

Journal: Nanomaterials (ISSN 2079-4991)

Manuscript ID: nanomaterials-2804311

Type: Article

Title: ADVANCED REMOVAL OF DYES THROUGH TUNING CARBON/TiO2 COMPOSITE PROPERTIES

Authors: Halyna Yankovych * , Coset Abreu Jauregui , Judit Farrando-Perez , Inna Melnyk , Miroslava Václavíková , Joaquín Silvestre-Albero *

Section:Energy and Catalysis

Special Issue: Applications of Nanomaterials for Electrocatalysis, Photocatalysis, Photoelectrochemical Solar Cells and Toxicity

The manuscript deals with preparation and characterisation of carbon-promoted composite for possibile application in dyes removal.

The authors report interesting results about the various composite they synthetised, with sufficient accuracy (figures, tables, referencing are clear and updated).

There is only one point that deserves a little clarification in my opinion: in the Raman characterisation section (pages 5 and 6 of the manuscript), they report several spectral components ascribed to the titania presence without a specific attribution to the possible titania polymorphs: this is in contrast with the results obtained by means of the XRD characterisation, in which all the three titania polymorphs are claimed to be present. It is necessary to homogenise the conclusions coming from both techniques in order to have (i) a clear figure of the strucuture of the variuos composites and (ii) the possible effect brought about by the presence of the different promoters added to the composites.

After this implementation (MINOR REVISIONS), the manuscript could be accepted for publication.

Comments on the Quality of English Language

Minor revisions (might be carried out even in the proofreading phase)

Author Response

Manuscript ID: nanomaterials-2804311

Type: Article

Title: ADVANCED REMOVAL OF DYES THROUGH TUNING CARBON/TiO2 COMPOSITE PROPERTIES

Authors: Halyna Yankovych * , Coset Abreu-Jauregui , Judit Farrando-Perez , Inna Melnyk , Miroslava Václavíková , Joaquín Silvestre-Albero *

Section: Energy and Catalysis

Special Issue: Applications of Nanomaterials for Electrocatalysis, Photocatalysis, Photoelectrochemical Solar Cells and Toxicity

The manuscript deals with preparation and characterisation of carbon-promoted composite for possibile application in dyes removal.

The authors report interesting results about the various composite they synthetised, with sufficient accuracy (figures, tables, referencing are clear and updated).

There is only one point that deserves a little clarification in my opinion: in the Raman characterisation section (pages 5 and 6 of the manuscript), they report several spectral components ascribed to the titania presence without a specific attribution to the possible titania polymorphs: this is in contrast with the results obtained by means of the XRD characterisation, in which all the three titania polymorphs are claimed to be present. It is necessary to homogenise the conclusions coming from both techniques in order to have (i) a clear figure of the strucuture of the variuos composites and (ii) the possible effect brought about by the presence of the different promoters added to the composites.

After this implementation (MINOR REVISIONS), the manuscript could be accepted for publication.

Answer: Thank You for the comment. Please, find below the explanation, which was added to the body of the manuscript and highlighted in yellow:

“Raman spectrometry was applied to the synthesized samples to identify the main vibrational modes within the composites (Figure 3). Raman spectrum of pristine activated carbon shows the two characteristic peaks at 1336 cm-1 and 1588 cm-1, corresponding to the D (sp3) and G (sp2) contributions, respectively. The presence of 2D and S3 peaks on the activated carbon Raman spectrum evidences the layered packing structure in these carbon materials [16]. Additionally, the ratio between intensities, ID/IG, determines the graphitization degree. This value is 1.06 for the pristine activated carbon [16]. Raman spectrum of the AC/TiO2 sample shows the characteristic D and G bands of the carbon and the characteristic peaks of TiOat 152 cm-1, 391 cm-1, 493 cm-1 and 624 cm-1 [17]. The peaks can be attributed to mixture of anatase, brookite and rutile (at 152 and 624 cm-1), anatase/brookite (at 391 cm-1) and anatase at 493 cm-1. Thus, the evaluated sample contains a mixture of three titania polymorphs. In the specific case of the doped samples, i.e., AC/TiO2_Zr, AC/TiO2_Ce and AC/TiO2_Cu, some shifts in the bands attributed to titania were noticed, together with the disappearance of the 2D and S3 peaks; however, the presence of all polymorphs was still observed. The mentioned observations could be attributed to the modification of the C-C vibrational modes upon modification with TiOand/or doped TiO2 but without modification of the graphitization degree. At this point, it is important to highlight that the Ti-O vibrational mode Eg(1) at 152 cm-1 remains unchanged.”

And in XRD part:

“X-ray diffraction patterns were used to investigate the crystalline structure of the synthesized materials. As demonstrated in Figure 4, the pristine activated carbon exhibits the characteristic peaks of amorphous carbon with broad peaks at 2θ: 22° and 44° and peaks attributed to quartz Q, with the most intensive patterns at 2θ: 20.9°, 26.7°, 50.2° (ICDD Card No. 00-046-1045). The XRD pattern of the composites shows the contributions from anatase (ICDD Card No. 21-1272), brookite (ICDD Card No. 29-1360) and rutile (ICDD Card No. 21-1276) with XRD patterns of anatase A at 2θ: 25.23° (101), 37.72° (004), 47.89° (200); rutile R at 2θ: 27.45° (110), 36.10° (101), 41.26° (111) and 54.35° (211), and brookite B at 2 θ: 25.35° (210), 25.71° (111), 30.83° (211) and 36.29° (102). The results coincide with the data from Raman spectroscopy, where the same composition of synthesized samples was detected. Overall, the XRD peaks due to TiO2 cannot be properly resolved due to the small TiO2 content in the composites and the proper dispersion (only some broad contributions due to anatase can be appreciated), while no sign for the dopants could be observed. It is assumed that these dopants could be integrated into the titania lattice without large changes in the titania crystalline structure [18]. At this point, it is important to highlight that the presence of anatase, brookite or rutile in the composites can boost the photocatalytic performance, particularly in the removal of non-biodegradable persistent pollutants [19].”